# Mathematical Modeling of the Effects of Temperature and Modified Atmosphere Packaging on the Growth Kinetics of Pseudomonas Lundensis and Shewanella Putrefaciens in Chilled Chicken

**DOI:** 10.3390/foods11182824

**Published:** 2022-09-13

**Authors:** Xutao Mai, Wenzhuo Wang, Xinxiao Zhang, Daoying Wang, Fang Liu, Zhilan Sun

**Affiliations:** 1Jiangsu Key Laboratory for Food Quality and Safety-State Key Laboratory Cultivation Base, Ministry of Science and Technology, Nanjing 210014, China; 2Institute of Agricultural Products Processing, Jiangsu Academy of Agricultural Sciences, Nanjing 210014, China; 3Key Laboratory of Cold Chain Logistics Technology for Agro-Product, Ministry of Agriculture and Rural Affairs, Nanjing 210014, China

**Keywords:** chilled chicken, *Pseudomonas lundensis*, *Shewanella putrefaciens*, modified atmosphere packaging, Gompertz, Ratkowsky

## Abstract

The effects of modified atmosphere packaging (MAP) on the growth and spoilage characteristics of *Pseudomonas lundensis* LD1 and *Shewanella putrefaciens* SP1 in chilled chicken at 0–10 °C were studied. MAP inhibited microbial growth, TVB-N synthesis, and lipid oxidation. The inhibitory effect of MAP became more significant as the temperature decreased. The kinetic models to describe the growth of *P. lundensis* LD1 and *S. putrefaciens* SP1 at 0–10 °C were also established to fit the primary model Gompertz and the secondary model Ratkowsky. The models had a high degree of fit to describe the growth of dominant spoilage bacteria in chilled chicken. The observed numbers of *P. lundensis* LD1 and *S. putrefaciens* SP1 at 2 °C were compared with the predicted numbers, and the accuracy factor and bias factor ranged from 0.93 to 1.14. These results indicated that the two models could help predict the growth of *P. lundensis* and *S. putrefaciens* in chilled chicken at 0–10 °C. The analyzed models provide fast and cost-effective alternatives to replace traditional culturing methods to assess the influence of temperature and MAP on the shelf life of meat.

## 1. Introduction

Chilling chicken can ensure its freshness to the greatest extent. Chilled chicken is highly popular because of its good tenderness and taste. However, the preservation of chilled chicken still faces many challenges [1]. High water activity and the high protein characteristics of chilled chicken provide favorable conditions for the growth of microorganisms, which can easily spoil chilled chicken. Among them, Pseudomonas and Shewanella are the most common dominant spoilage bacteria during the later stage of storage because they can tolerate low storage temperatures and rapidly grow during storage [2]. *Pseudomonas* is a psychotrophic aerobic bacterium with a strong metabolic capacity, which can hydrolyze the protein and lipolysis of chilled chicken to produce harmful volatiles and can grow under modified atmosphere packaging (MAP) [3]. *Shewanella* is another major spoilage bacterium in chilled chicken. It belongs to the typical psychrophilic facultative anaerobe, which can produce hydrogen sulfide, hydrolyze extracellular enzymes, and accelerate the spoilage of high-protein chilled chicken, thereby shortening its shelf life.

Modified atmosphere packaging (MAP) has been increasingly utilized in the storage and preservation of meat and meat products [4]. Currently, modified atmosphere packaging (MAP) combined with refrigeration is increasingly used for the storage and preservation of meat and meat products. The optimum gas environment for chilled chicken preservation is 80% N_2_ and 20% CO_2_. CO_2_ inhibits the respiration of aerobic bacteria and reduces their growth in meat by inhibiting their metabolism and cell division, thereby preventing meat spoilage [5]. However, modified atmosphere packaging has also obvious shortcomings. Excess carbon dioxide can cause the package to collapse. However, a less amount of CO_2_ such as 20% CO_2_/80% N_2_ does not completely inhibit the growth of microorganisms. Studies have reported that despite the high carbon dioxide levels and no residual oxygen measured in modified atmosphere packaging (MAP), vigorous growth of Pseudomonas can still be found in meat [3].

MAP combined with low temperatures is a more common way to preserve chilled meat [6]. The commonly used storage temperature is 0–4 °C. According to Guerra Monteiro et al., MAP with a gas composition of 60% N_2_ and 40% CO_2_ could prolong the freshness of tilapia fillets by 9 days at 4 °C. MAP is a very promising technology [7]. Jiménez et al. found that, in samples stored at 4 °C, MAP extended shelf life to 21 days, compared with 5 days for NP [8]. However, temperature fluctuations often occur at different stages of the food supply chain, including storage, transportation, and sale. Especially during the transportation and sale stage, it is difficult to keep the storage temperature at 0–4 °C all the time due to defective refrigeration equipment. It will frequently fluctuate within a certain temperature range (0–10 °C), which will cause the storage of chilled chicken to deviate from the standard or recommended temperature, and have a cumulative adverse effect on the quality of stored food, thereby accelerating the deterioration of chilled chicken [9]. Compared with MAP, storage temperature fluctuations have the greatest impact on microbial growth kinetics. Studies have found that a sharp decrease in shelf life can be observed in foods stored at variable temperatures compared with those stored at constant temperatures, which is a major cause of food damage.

However, it is not unrealistic to perform a specific analysis of the microbial counts of each batch of chilled chicken during transportation and sale. Understanding the dynamics of microbial populations within a certain temperature range during storage is important to maintain the quality and safety of frozen chicken. To this end, a dynamic model should be developed to elucidate the effect of variable temperature on major spoilage bacteria and predict the shelf life of chilled chicken [10,11,12]. At present, the modified Gompertz model has shown great potential for the dynamic prediction of microorganisms in food. It has the advantages of fast detection speed, high efficiency, and economic convenience. Li et al. successfully confirmed its superior performance in the growth of Pseudomonas [13].

This study first evaluated the potential of MAP and low temperature to control the growth of two dominant spoilage bacteria (*Pseudomonas lundensis* LD1 and *Shewanella putrefaciens* SP1). The growth kinetics of the two analyzed strains were established under NP and MAP at 0–10 °C. Gompertz was selected to analyze the growth data of the two strains at 0 °C, 4 °C, and 10 °C under NP and MAP. The effects of environmental factors on the parameters of the first-order model were described using the Ratkowsky model as a second-order model. The growth data (Af and Bf) at 2 °C were used to validate the established model. The purpose of this study was to predict the shelf life of chilled chicken under NP and MAP in the range of 0–10 °C.

## 2. Materials and Methods

### 2.1. Preparation of Bacterial Suspension

*P. lundensis* LD1 and *S. putrefaciens* SP1 were preserved in the laboratory. These two strains were inoculated into BHI broth and incubated at 30 °C in a shaker (200 rpm) overnight. The cultures were transferred to fresh media for further culture until OD_600_ reached 1.0. Subsequently, 5 mL bacterial cultures were collected via centrifugation (8000× *g*, 3 min), washed with physiological saline (PS), and resuspended in 5 mL of PS. The bacterial cultures were diluted to the correct concentration for later use.

### 2.2. Sample Preparation and Inoculation

Common fresh chicken breasts were harvested from a supermarket in Nanjing, China. The minced chicken breasts were divided into small portions of 20 g. All the samples were sterilized via cobalt-60r radiation treatment (18 kGy). The above-activated bacteria were inoculated into sterilized samples at a final concentration of 3–4 lg CFU/g. NP (air) and MAP (N_2_/CO_2_, 80%/20%) were then prepared. Both MAP and NP are encapsulated using polypropylene films. The samples were kept at 0 °C, 2 °C, 4 °C, 8 °C, and 10 °C for different lengths of time, depending on their storage temperature. The samples were analyzed every 2 days at 8 °C and 10 °C, followed by every 3 days at 0 °C, 2 °C, and 4 °C. At least three samples in NP and MAP were randomly selected from subsequent microbial spoilage in chicken meat.

### 2.3. Microbiological Analysis

About 20 g of minced chicken meat was homogenized in sterile blender bags with 180 mL of PS for 1 min to detach the microorganisms from the chicken breast. The buffer was transferred from each of the bags into sterile 10 mL centrifuge tubes for 10 times gradient dilution. The viable counts of *P. lundensis* LD1 and *S. putrefaciens* SP1 were determined using appropriate dilutions on LB agar.

### 2.4. Detection of Corruption Characteristics

#### 2.4.1. Total Volatile Basic Nitrogen (TVB-N) Assay

The chicken breast samples were mixed with distilled water and homogenized to uniformly disperse the sample solution. The samples were then filtered for further measurement. The TNB-N was analyzed according to the national GB micro-diffusion method standard (GB 5009.228-2016).

#### 2.4.2. Lipid Oxidation

The chicken breast was divided into the corresponding number of samples with 5 g as the standard. Thiobarbituric acid-reactive substances (TBARS) were selected to determine lipid oxidation through the procedures used in national standards (GB 5009.181-2016).

### 2.5. Growth Kinetics and Mathematical Modeling

#### 2.5.1. Primary Models

In this study, a modified Gompertz model was selected to match the initial data of the growth of *P. lundensis* LD1 and *S. putrefaciens* SP1 in chicken meat under the storage conditions of 0–10 °C.

This sigmoid function model can accurately describe the relationship between time and the growth of microorganisms under constant environmental conditions, and the parameters of the modified Gompertz model are physiologically meaningful.

The modified Gompertz model equation is as follows:$$P\;(t)\;=p_{0}+\;(p_{\max}-p_{0})\times \;\exp\;\{-\exp\;[\frac{\mu_{\max}e}{p_{\max-p_{0}}}\;\left(\lambda -t\right)+1]\}$$

In the above formula, λ is the lag time (h); µ_max_ represents the maximum specific growth rate of microbial growth (lg CFU/g/h); P (t) is the number of viable bacteria at a certain time t (lg CFU/g); p_0_ and p_max_ are the initial and maximum numbers of viable bacteria (lg CFU/g), respectively.

The performance of the primary model used in this study was evaluated by obtaining the MSE using the following equation:$$\mathrm{MSE}=\sum_{i=1}^{n}\frac{\;{\left(obs-pred\right)}^{2}}{n-q}$$
where *obs* is the observed value, *pred* is the predicted value, *q* is the number of model parameters, and *n* is the number of observations.

#### 2.5.2. Secondary Models

The Ratkowsky model was used to visually describe the effect of the temperature on the growth kinetic parameters of *P. lundensis* LD1 and *S. putrefaciens* SP1 in the first-level model.

The Ratkowsky model equation is as follows:$$\sqrt{\mu_{\max}}=b_{µ\max}\times (T-T_{0})\;\frac{1}{\sqrt{\lambda }}\;=b_{\frac{1}{\sqrt{\lambda }}}\;\times \;(T-T_{1})$$

In the above formula, T is the storage temperature; T_0_ and T_1_ are the theoretical minimum growth temperatures of dominant spoilage bacteria, that is, µ_max_ at these temperatures is 0; b_µmax_ and $b_{\frac{1}{\sqrt{\lambda }}}$ are the parameters of the equation.

#### 2.5.3. Validation of the Predictive Models

This study used the accuracy factor (A_f_) and the bias factor (B_f_) to estimate the performance of the predictive model. A_f_ is an index used to evaluate the prediction accuracy of the prediction model, and B_f_ is an index used to evaluate the difference between the predicted value and the measured value. The equations for A_f_ and B_f_ are as follows:$$A_{f}=10^{[\sum \left|pred-obs\right|]/n}$$
$$B_{f}=10^{[\sum \left(pred-obs\right)]/n}$$
where *pred*, *obs*, and *n* are the predicted value, the observed value, and the number of repetitions of the observed data, respectively.

### 2.6. Curve Fitting and Statistical Analysis

SPSS 11.0 (IBM, Armonk, NY, USA) was used for statistical analysis. The viable counts were determined using the natural logarithm (lg) of dominant spoilage bacteria. MATLAB2019B (fitting toolbox) and Origin 8.5 were used for modeling and simulation. This method could quickly and accurately predict the growth of dominant spoilage bacteria in chilled chicken at different temperatures through a combination of data and charts. The goodness of fit was assessed using the coefficient of determination (R^2^), whereas the performance of the model was analyzed using the mean squared error (MSE).

## 3. Results and Discussion

### 3.1. Microbiological Analyses of Chilled Chicken

As dominant spoilage bacteria, the numbers of *P. lundensis* LD1 and *S. putrefaciens* SP1 are important indicators to determine the quality of chilled chicken. As shown in Figure 1, the initial count of *P. lundensis* LD1 inoculated into the chilled chicken samples was 3.86 lg CFU/g. After 6 days of storage at 0 °C, the viable counts in the NP group reached 7.28 lg CFU/g, which exceeded the national limit of microorganisms in chilled meat (7 lg CFU/g) [1]. However, the number in the MAP group was only 4.69 lg CFU/g, which still met the edible standard. In addition, changes in the number of *S. putrefaciens* SP1 in the chilled chicken samples at 0 °C were monitored. On the 9th day, the number of *S. putrefaciens* SP2 in the NP group reached 7.12 lg CFU/g. In the MAP group, similar to *P. lundensis* LD1, a final concentration of *S. putrefaciens* SP1 reached 6.37 lg CFU/g after 9 days of storage. Thus, the growth of *S. putrefaciens* SP2 was slower than that of *P. lundensis* LD1. This might be due to the fact that *P. lundensis* LD1 is a psychotrophic bacterium that can adapt to a low-temperature environment.

At 4 °C, the viable count of *P. lundensis* LD1 in the NP group was observed to be 8.5 lg CFU/g on day 4, which far exceeded the acceptable level of 7 lg CFU/g [1]. At the end of storage on day 8, the maximum viable count of *P. lundensis* LD1 in the MAP group was 7.83 lg CFU/g. The growth rate of *P. lundensis* LD1 in the MAP group was slower than that in the NP group. This result indicated that MAP inhibited the growth of *P. lundensis* LD1. Over the 8-day storage time, the bacterial concentration of *S. putrefaciens* SP1 in the NP group gradually but significantly increased. The final concentration of *S. putrefaciens* SP1 was observed to be close to 8 lg CFU/g after storage of up to 8 days. For the MAP group, the viable count was found to be only 6.49 lg CFU/g. Thus, MAP treatment at 4 °C had a positive effect on preventing the growth of *S. putrefaciens* SP1. The growth of the two dominant spoilage bacteria at 10 °C was different from that at low temperatures. With the increase in storage temperature, the two strains in the NP group grew rapidly, and they were more than 7 lg CFU/g after 2 days of storage in all the samples, which exceeded an acceptable upper limit for fresh meat. They were no longer edible, and the final population of the two strains was close to 10 lg CFU/g. MAP had little effect on the number of the two dominant bacteria at 10 °C. The higher the temperature, the faster the respiration rate of *P. lundensis* LD1 and *S. putrefaciens* SP1, and the weaker the antibacterial effect of MAP.

Research on reducing meat spoilage remains a formidable challenge. Controlling the storage temperature and MAP are the two most important fresh-keeping methods for chilled meat. The temperature is a key factor affecting the quality of meat. Appropriately lowering the temperature can slow down the rate of chemical reactions caused by microorganisms and inhibit the activity of metabolic enzymes, thereby prolonging the shelf life. Therefore, low-temperature storage [14] is the most commonly used and most effective means of preservation. Near-freezing temperature storage (−2 °C) has been proposed and verified in recent years to have a better preservation effect on chilled meat compared with traditional chilling at 4 °C. In Asia, the shelf-life extension of frozen yellow feather meat often relies on near-freezing storage (−2 °C) [15]. Studies have found that this technology has a significant effect on maintaining the nutritional, textural, and sensory qualities of meat, thereby inhibiting microbial growth and hindering harmful chemical changes. Although low-temperature fresh-keeping technology has greatly advanced the development of fresh-keeping methods, some of its defects cannot be ignored. Given the shortcomings of a single low-temperature preservation method, the current research focused on several innovative and effectively combined technologies. For example, the combination of low temperature and MAP technology has been explored and successfully applied to chilled meat.

### 3.2. Analysis of Spoilage Characteristics

#### 3.2.1. TVB-N during Storage

Chilled chicken contains a large amount of protein. Due to the microbial and endogenous enzyme activity, protein is degraded, resulting in the production of TVB-N. The accumulation of TVB-N can make the surface of chilled chicken moist and sticky, inelastic, and emit a bad smell. Thus, TVB-N is one of the evaluation indexes of meat corruption [16]. Figure 2 displays the TVB-N value in the different packaging methods. After 9 days of storage at 0 °C, the TVB-N values in the chicken samples of the NP and MAP groups inoculated with *P. lundensis* LD1 were 42 and 33.25 mg/100 g, respectively. Compared with the NP group, the TVB-N value of the MAP group decreased by 8.75 mg/100 g (*p* < 0.05). the TVB-N of the NP group inoculated with *S. putrefaciens* SP1 was 40 mg/100 g, which was 9.7 mg/100 g higher than that of the MAP group (*p* < 0.05).

Previous studies reported that MAP with high concentrations of carbon dioxide helps inhibit the growth of microorganisms that produce volatile compounds, such as Gram-negative aerobic bacterium *Pseudomonas* [17]. At different storage temperatures, the TVB-N of the MAP group was lower than that of the NP group. On the basis of the acceptability of human senses (taste) and the analysis of chicken spoilage and freshness, for the chicken samples stored in the air, a TVB-N value of 40 mg/100 g has been recommended as the upper limit of chilled chicken spoilage and shelf life indicators [18]. The results of this study revealed that MAP and low temperatures had a particularly remarkable effect on the TVB-N value. For example, the TVB-N of the chilled chicken samples inoculated with *P. lundensis* LD1 reached 33.8 mg/100 g when stored at 10 °C for 4 days. It took 6 days to reach the same TVB-N value at 0 °C, possibly because a higher storage temperature facilitates the enhancement of enzyme activity and microbial metabolism. However, this rapid accumulation of TVB-N was less affected by MAP with the increase in temperature. Microbial growth is also affected by low temperatures; low microbial metabolism leads to reduced protein and fat degradation. Endogenous enzyme activity is also inhibited at low temperatures, reducing protein breakdown. The observation of low-temperature bacteriostasis was consistent with the results reported in the literature [19]. MAP could also reduce TVB-N production, which might be mainly achieved by reducing microbial growth. At high temperatures, microbial growth was accelerated, and MAP had less inhibitory effect on microorganisms.

#### 3.2.2. Lipid Oxidation

Lipid oxidation is one of the key factors in the deterioration of meat and meat products. Lipid oxidation is mainly a chemical reaction that occurs under aerobic conditions and is affected by factors such as lipoxygenase, oxygen, temperature, and light in chilled chicken. A series of oxidation reactions can occur during storage, the most important of which is the potential response of polyunsaturated fats to oxygen, and the reaction can lead to the accumulation of malondialdehyde (MDA) [20,21].

As shown in Figure 3, after 6 days of storage at 0 °C, the TBARS content of the NP group inoculated with *P. lundensis* LD1 was 0.66 mg MDA/kg. After 4 days of storage at 10 °C, 0.76 mg MDA/kg was detected. When the temperature increased, the absorption of oxygen by free radicals was enhanced, and the enzyme activity improved, thereby accelerating lipid oxidation. At 0 °C, 4 °C, and 10 °C, the MAP group inoculated with *P. lundensis* LD1 had 0.6, 0.39, and 0.45 mg MDA/kg on days 6, 4, and 2, respectively. The TBARS content slowly increased in the MAP group. The absence of oxygen in the MAP group reduced the oxygen uptake by meat and inhibited the activity of endogenous enzymes. Therefore, these factors slowed down the oxidation of highly unsaturated fatty acids to a certain extent. Similar to *P. lundensis* LD1, MAP also reduced the TBARS value in the NP group inoculated with *S. putrefaciens* SP1. Similar conclusions were drawn by Pongsetkul et al., who used dry fermented catfish as raw material and found that indicators such as oxidation products (PV and TBARS values) and sensory acceptability vary slowly under MAP and VP conditions. Shelf life was extended from 1–2 months at room temperature to 90 days [22]. Demirhan et al. also found that MAP inhibits microbial growth and delayed lipid and protein oxidation [23]. Thomas et al. proved that MAP protects the quality of chicken and extends its shelf life [24]. Chilled chicken contains a large amount of unsaturated fatty acids, which are highly sensitive to oxygen, thereby causing a reaction to generate free radicals and lipid peroxyl radicals. Lipid oxidation produces off-flavors and leads to rancid meat by breaking down lipid hydroperoxides into volatile aldehydes and ketones [25]. Chilled chicken can be protected from oxidation by the use of MAP technology and low-temperature storage conditions.

### 3.3. Mathematical Modeling

#### 3.3.1. Primary Models

The growth data for *P. lundensis* LD1 and *S. putrefaciens* SP1 were obtained for the chilled chicken samples stored at 0–10 °C, equipped with a modified Gompertz model (Figure 4). With the increase in culture temperature, the retention period of dominant spoilage bacteria was shortened, and the maximum specific growth rate was accelerated. All models exhibited typical sigmoid shapes. As shown in Table 1, the lag period of the NP group inoculated with *P. lundensis* LD1 at 10 °C was 7.56, which was much lower than 39.28 at 0 °C. Notably, the changing rule of µmax showed the opposite trend. The µmax value of 0.14 at 10 °C was much higher than 0.035 at 0 °C. However, the NP groups inoculated with *S. putrefaciens* SP1 at 10 °C and 0 °C were fitted with the λ values of 8.99 and 45.93, respectively. At 0 °C, the maximum specific growth rate was 0.022, which meant that the microorganisms grew extremely slowly. Once raised to 10 °C, µmax was as high as 0.1, which was almost five times higher than at 0 °C. The enzyme activity increased at 10 °C, thereby accelerating the growth of microorganisms. When the storage temperature was constant, the lag period of the NP group of the two dominant spoilage bacteria was shorter than that of the MAP group, and the specific growth rate of the NP group was much greater than that of the MAP group. For example, at 4 °C, the lag phase of the NP group inoculated with *P. lundensis* LD1 was 19.35, and the maximum specific growth rate was 0.08. However, the lag period of the MAP group was prolonged by 7.67, and the specific growth rate was slowed down by 0.05. Similar trends were observed at other temperatures. These results on µmax and λ at different storage temperatures were consistent with the previous work [26].

At 10 °C, the chilled chicken samples inoculated with *P. lundensis* LD1 or *S. putrefaciens* SP1 had a shelf life of only one day in NP. With the decrease in storage temperature, the shelf life was prolonged to varying degrees. The shelf life of the NP samples inoculated with *P. lundensis* LD1 at 0 °C was 5 days, while that of *S. putrefaciens* SP1 was extended from 1 day to 9 days. For the chilled chicken samples packaged in MAP, the samples containing *P. lundensis* LD1 or *S. putrefaciens* SP1 had a shelf life of 2 days at 10 °C. With the decrease in storage temperature, the storage effect of samples using the MAP method was obvious. The shelf life of the samples inoculated with *P. lundensis* LD1 or *S. putrefaciens* SP1 were 12 and 25 days, respectively, which were 7 and 16 days longer than that of the samples stored using NP, respectively.

In this study, the MSE for fitting the Gompertz model ranged from 0.000451 to 0.0897, with R^2^ values above 0.984. MSE determines the error size of the model; the closer it is to 0, the more accurate the model is. By analyzing the corresponding MSE and R^2^ values at different temperatures, the Gompertz model had high goodness of fit for the number of the two dominant spoilage bacteria at all storage temperatures. The coefficients of determination were all greater than 0.985. In particular, the growth of *P. lundensis* LD1 and *S. putrefaciens* SP1 in the MAP group had a low MSE and high R^2^; R^2^ was greater than 0.99, and the maximum value of MSE was only 0.073. Thus, the modified Gompertz model was suitable for simulating the growth of dominant spoilage bacteria in the chilled chicken samples. The modified Gompertz model has been successfully used to predict the growth of Pseudomonas in pork, and a small MSE of 0.33 and a large R^2^ of 0.98300.0041 were obtained [13]. Gholllasi-Mood et al. used the Gompertz model to predict the growth of a specific spoilage organism Pseudomonas in air-packed chicken stored at different temperatures with satisfactory results. The R^2^ value was greater than 0.98, and the MSE value was less than 0.18 [27].

#### 3.3.2. Secondary Model

The above-modified Gompertz model could predict the growth of dominant spoilage bacteria at 0–10 °C, but it could not describe the effect of temperature on the number of dominant spoilage bacteria. The fresh-keeping temperature of chilled chicken during storage could only be maintained within a temperature range. Therefore, a second model was necessary. The Ratkowsky model was used to fit µmax and λ derived from the Gompertz model [28,29]. As shown in Figure 5, a good linear relationship was found between the temperature and the square root of µmax and the inverse of the square root of λ. The R^2^ values of $\sqrt{µ\max}$ and $\sqrt{1/\lambda }$ of *P. lundensis* LD1 were 0.943 and 0.983 under NP, corresponding to the MSE values of 0.01883 and 0.02089. Under MAP, they were 0.98 and 0.99, respectively, while the MSE values were 0.01018 and 0.0167. The R^2^ values corresponding to *S. putrefaciens* SP1 were all above 0.96, and the MSE values were all less than 0.01988. The lag phase and exponential growth phase play extremely important roles in microbial growth. Microorganisms generally do not grow immediately in the lag period. They live in a state of equilibrium, in which the number of cells maintains a dynamic balance; thus, the chilled chicken samples did not deteriorate during this period. The microorganisms in the exponential growth period have a strong metabolic capacity, which has a great impact on the quality of chilled chicken. µmax and λ can show the changes in dominant spoilage bacteria and have guiding roles in ensuring food safety. The µmax and λ of dominant spoilage bacteria in chilled chicken meat are most affected by temperature [30]. As shown in Figure 5, µmax increased with the temperature, whereas λ decreased with the increasing temperature. At 8–10 °C, significant growth was observed in terms of µmax and λ. When the temperature rose to the optimum growth temperature range of the dominant spoilage bacteria, the intracellular enzymes became active, and the metabolic rate increased, indicating rapid multiplication.

µ_max_ in the NP group was greater than that in the MAP group, whereas the λ value in the NP group was smaller than that in the MAP group. CO_2_ in the MAP group possibly inhibited the growth of *P. lundensis* LD1 and *S. putrefaciens* SP1. However, the inhibitory effect of MAP on *P. lundensis* LD1 and *S. putrefaciens* SP1 was less effective when the temperature increased. In the MAP group for the two dominant spoilage bacteria, the lower the temperature, the smaller the changes in λ and µ_max_. When stored at low temperatures, the temperature was the most important factor to inhibit the growth of the strain. When the temperature increased, the growth of microorganisms was accelerated, and MAP became the main factor to inhibit the growth of the strains.

### 3.4. Model Validation

The applicability of the modified Gompertz model was verified through independent experiments. The observed values were compared with the predicted values in the experiments, and the models were mathematically evaluated using B_f_, A_f_, and MSE [10].

The bacterial concentrations of *P. lundensis* LD1 and *S. putrefaciens* SP1 in the chilled chicken samples were measured at 2 °C with a temperature range of 0–10 °C to test and validate the performance of the predictive model. The obtained data were compared with the predicted data of *P. lundensis* LD1 and *S. putrefaciens* SP1 (Figure 6). No significant difference was observed. The model could roughly determine the total number of colonies at a range of 0–10 °C.

## 4. Conclusions

In our study, the effects of temperature and MAP on the microorganisms, TVB-N, and lipid oxidation of chilled chicken were analyzed. Results showed that low temperature and MAP inhibited microbial growth, decreased TVB-N synthesis, and inhibited lipid oxidation. To a certain extent, the quality of the product was protected. Compared with NP, MAP had a longer microbial retention period and a lower growth rate. At the end of storage, the number of dominant spoilage bacteria detected in MAP was less than that in NP. However, it could not completely inhibit the growth of dominant spoilage bacteria. After the above analysis, a model could be established to reasonably grasp the effect of MAP on the quality of chilled chicken.

The improved Gompertz and Ratkowsky models were selected to analyze the primary and secondary models of the two strains. MSE and R^2^ confirmed that the models were reliable. According to λ and µ_max_, the temperature was the most important factor to inhibit the growth of the strains. The model predicted the growth of *P. lundensis* LD1 and *S. putrefaciens* SP1 at 2 °C, and A_f_ and B_f_ showed strong fitting ability. We constructed this model to predict the growth of *P. lundensis* LD1 and *S. putrefaciens* SP1 in chilled chicken at 0–10 °C. This study provides valuable information on the growth kinetics of *P. lundensis* LD1 and *S. putrefaciens* SP1. It also lays the foundation for the growth prediction of natural microflora in chilled chicken.

## Figures and Tables

**Figure 1 foods-11-02824-f001:**
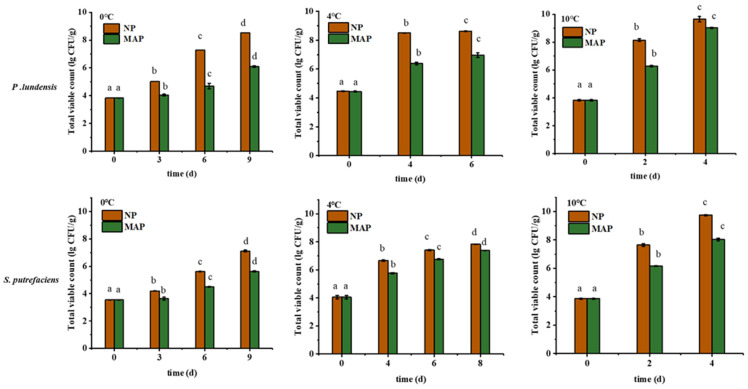
Growth of *P. lundensis* and *S. putrefaciens* in chilled chicken at 0 °C, 4 °C, and 10 °C. Lowercase letters: significance of dominant spoilage bacteria at different storage times.

**Figure 2 foods-11-02824-f002:**
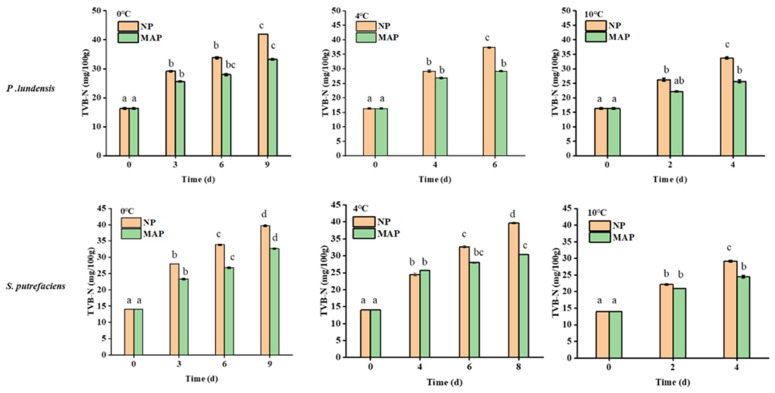
TVB-N content values of chicken breast samples for different treatment groups. Lowercase letters: significance of dominant spoilage bacteria at different storage times.

**Figure 3 foods-11-02824-f003:**
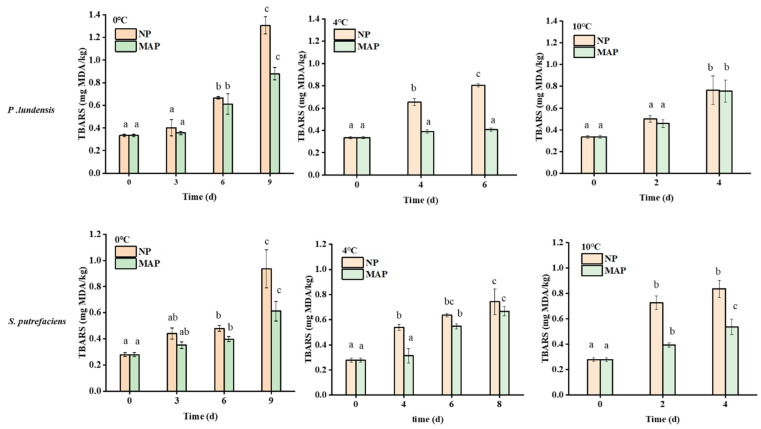
Effect of normal packaging (NP) and modified atmosphere packaging (MAP) on the TBARS values of chicken breast meat. Lowercase letters: significance of dominant spoilage bacteria at different storage times.

**Figure 4 foods-11-02824-f004:**
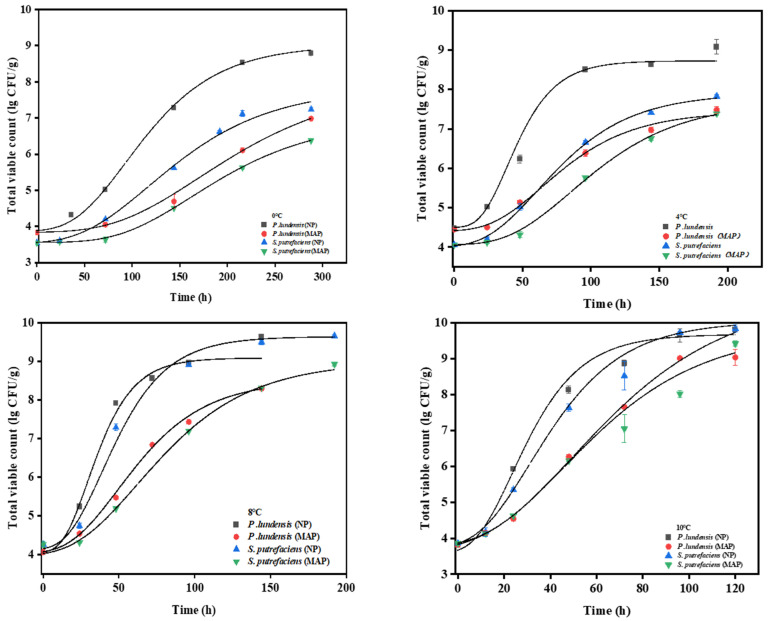
Growth kinetic model curves of two dominant spoilage bacteria in chilled chicken at 0 °C, 4 °C, 8 °C, and 10 °C using the modified Gompertz equation.

**Figure 5 foods-11-02824-f005:**
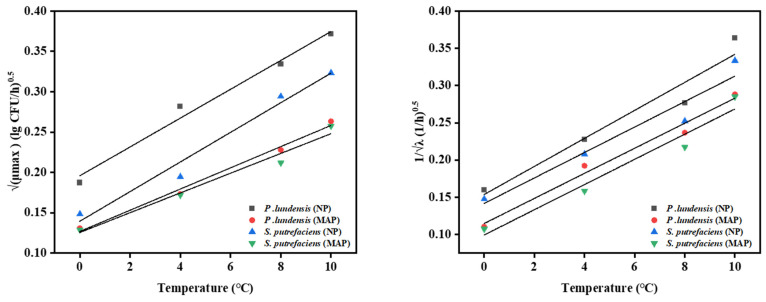
Maximum specific growth rates obtained and lag phase durations from the modified Gompertz model fitted to Ratkowsky.

**Figure 6 foods-11-02824-f006:**
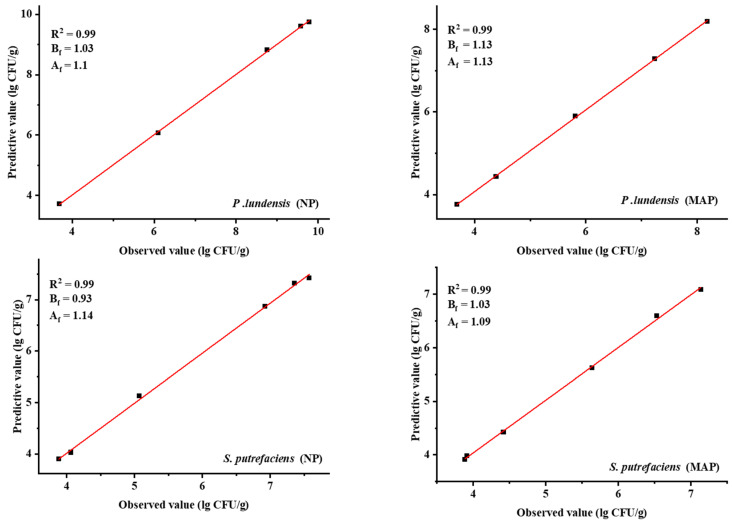
Comparison of the observed and predicted numbers of dominant spoilage bacteria in chilled chicken samples stored at 2 °C under different packaging methods.

**Table 1 foods-11-02824-t001:** Fitting parameters of the primary growth models.

	Temperature (°C)	P_0_ (lg CFU/g)	P_max_ (lg CFU/g)	µ_max_ (h^−1^)	λ (h)	MSE	Pseudo-R^2^
*P. lundensis* (NP)	0 °C	3.86	9	0.0351	39.28	0.0653	0.999
	4 °C	4.5	8.72	0.0796	19.35	0.0653	0.996
	8 °C	4.04	9.08	0.1115	13.06	0.0673	0.996
	10 °C	3.58	9.68	0.1381	7.56	0.0616	0.984
*P. lundensis* (MAP)	0 °C	3.83	7.94	0.0171	82.07	0.00824	0.999
	4 °C	4.4	7.44	0.03	27.02	0.00833	0.986
	8 °C	4.02	8.5	0.0519	17.85	0.00712	0.996
	10 °C	3.65	10.77	0.0693	12.04	0.073	0.995
*S. putrefaciens* (NP)	0 °C	3.52	7.82	0.022	46.1	0.00091	0.999
	4 °C	4	7.89	0.0379	23.13	0.00357	0.996
	8 °C	4.11	9.63	0.0866	15.72	0.0467	0.985
	10 °C	3.74	10.05	0.1045	8.99	0.054	0.999
*S. putrefaciens* (MAP)	0 °C	3.55	6.97	0.0166	86.72	0.00045	0.999
	4 °C	4.05	7.69	0.0296	39.86	0.00324	0.998
	8 °C	3.96	8.99	0.0449	21.17	0.0146	0.999
	10 °C	3.74	9.08	0.0663	12.31	0.0897	0.99

## Data Availability

The data presented in this study are available on request from the corresponding author.

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
