# Peer review of "Mathematical Modeling of the Effects of Temperature and Modified Atmosphere Packaging on the Growth Kinetics of Pseudomonas Lundensis and Shewanella Putrefaciens in Chilled Chicken"

_foods, 2022, doi:10.3390/foods11182824_

Round 1
Reviewer 1 Report
1. In the introduction section
The introduction is concise and effictively gives a background and the aim is clear.
Line 35: "It is easy to breed a large number of microorganisms."........please rephrase and do not use "breed".
2. The material and methods
Described well the methodology.
Line 96: Please justify the use of final concentration of 3–4 lg CFU/g.
3. The results and discussion
Well written
please add significance values in the results section.
Author Response
Response to the reviewer #1’s comments:
Comment:1. Line 35: "It is easy to breed a large number of microorganisms."........please rephrase and do not use "breed".
Response: Thank you for your valuable suggestions. The Sentence has been revised. (Line35)
Comment: 2. Line 96: Please justify the use of final concentration of 3–4 lg CFU/g.
Response: Thank you for your valuable suggestions. The total number of colonies of fresh chicken in the market is 3-4lg CFU/g. To simulate the initial strain in chicken, the final concentration of 3-4 lg CFU/g was added in this paper. A similar study was conducted by Hilgarth et al., and the initial colony counts in the meat were all 3-4 lg CFU/g. You can see the following documents. Thanks again for your suggestions.
- Hilgarth, M.; Lehner, E.M.; Behr, J.; Vogel, R.F. Diversity and anaerobic growth of Pseudomonas spp. isolated from modified atmosphere packaged minced beef. Journal of Applied Microbiology 2019, 127, 159-174, doi:10.1111/jam.14249.
- Hilgarth, M.; Behr, J.; Vogel, R.F. Monitoring of spoilage-associated microbiota on modified atmosphere packaged beef and differentiation of psychrophilic and psychrotrophic strains. Journal of Applied Microbiology 2018, 124, 740-753, doi:10.1111/jam.13669.
Comment: 3. please add significance values in the results section.
Response: Thanks for your valuable comments. We added the significant difference between the different storage time of the dominant spoilage bacteria in Figure 1-3. The curves of the primary model and the data in Table 1 are obtained by fitting the original data. The data is the predicted value obtained from the model. We also see other literatures on constructing mathematical models, which also do not perform significant difference analysis in primary model (Li et al., 2013; Tarlak et al., 2017). Thanks again for your valuable opinion.
- Tarlak, F.; Ozdemir, M.; Melikoglu, M. Mathematical modelling of temperature effect on growth kinetics of Pseudomonas spp. on sliced mushroom (Agaricus bisporus). International Journal of Food Microbiology 2018, 266, 274-281, doi:10.1016/j.ijfoodmicro.2017.12.017.
- Li, M.; Niu, H.; Zhao, G.; Tian, L.; Huang, X.; Zhang, J.; Tian, W.; Zhang, Q. Analysis of mathematical models of Pseudomonas spp. growth in pallet-package pork stored at different temperatures. Meat Science 2013, 93, 855-864, doi:10.1016/j.meatsci.2012.11.048.
Reviewer 2 Report
The authors have developed a prediction model comprising the effect of the temperature of chicken stored under a modified atmosphere and under air. The temperature range was restricted to typical storage temperatures starting at 0°C to 10°C that might happen under transport conditions. In addition, they investigated the effect of microbial activity through measuring production of volatile basic nitrogen (protein degradation) and lipid oxidation within the same temperature range. As test microorganisms, Ps. lundensis and S. putrefaciens, were well chosen. The former is a psychrotroph (temperature range 0-37°C), and the latter is a facultative anaerobic mesophile (growth temperature range: 4-40°). All experiments, applied methodologies and data analysis were correctly performed, including the statistical analysis. The model is very useful as a prediction tool for storing chilled chicken. It is thus a very complete study, but there are issues to improve the manuscript.
Comments
1. Line 38 and elsewhere in the text: Pseudomonas ludensis is a psychotrophic bacterium (not a psychrophilic). It is known indeed as an aerobic bacterium, but anaerobic growth under MAP conditions has been documented (e.g. Hilgarth et al 2019 J Appl Microbiol 127:159-74). The authors should correct and update this part.
2. Line 53-54 reads like an odd sentence. Please rephrase, like e.g.: Even though small amounts of oxygen remain, due to the quality of the applied technology, growth of (facultative) anaerobes occurred.
3. Line 64-66 needs a reference.
4. In the last paragraph of the Introduction the Ratkowsky model deserves to be mentioned as well.
5. Line 134. Replace ‘MES’ by ‘MSE’ in the equation.
6. Line 158: lg10 is not the ‘natural logarithm’.
7. Overall comment ‘Results and Discussion’ section. When reading this section, it gets the impression that the ‘Results section’ and the ‘Discussion section’ can best be separated. In the current state the flow of the results in the different parts is disturbed by a discussion until the next results part starts. Each result section is a standalone and when combined will be more easily readable. It also prevents any repeats in the various discussion parts.
Secondly, this whole section can be written much more concisely. There are too many repeats, and many details are mentioned in the text that are already given in the figures.
Therefore, I advise the authors to improve this part of their manuscript.
8. Figures 1-4. Make sure that all y-axes are the same in each figure. For instance, in Figure 1 the second top panel (Ps. ludensis at 4°C) the Y-axis should also run to 10, like in the other panels.
9. Line 257: Write MDA full out as it is used here for the first time.
Explain here as well whether lipid oxidation is a microbial process or can it also be chemical oxidation in the presence of oxygen?
10. Line 356 Replace ‘weakened’ by ‘was less effective’.
11. Line 395 Typo, should be ‘provided’.
Author Response
Response to the reviewer #2’s comments:
Comment: 1. Line 38 and elsewhere in the text: Pseudomonas ludensis is a psychotrophic bacterium (not a psychrophilic). It is known indeed as an aerobic bacterium, but anaerobic growth under MAP conditions has been documented (e.g. Hilgarth et al 2019 J Appl Microbiol 127:159-74). The authors should correct and update this part.
Response: Sorry for such an error, we have added the corresponding reference. A revision of the corresponding position has been made. (Line 38-40,185-186).
- Hilgarth, M.; Lehner, E.M.; Behr, J.; Vogel, R.F. Diversity and anaerobic growth of Pseudomonas spp. isolated from modified atmosphere packaged minced beef. Journal of Applied Microbiology 2019, 127, 159-174, doi:10.1111/jam.14249.
- Doulgeraki, A.I.; Nychas, G.-J.E. Monitoring the succession of the biota grown on a selective medium for pseudomonads during storage of minced beef with molecular-based methods. Food Microbiology 2013, 34, 62-69, doi:10.1016/j.fm.2012.11.017.
Comment: 2. Line 53-54 reads like an odd sentence. Please rephrase, like e.g.: Even though small amounts of oxygen remain, due to the quality of the applied technology, growth of (facultative) anaerobes occurred.
Response: Thank you for your comment. The content of this section has been modified revised. (Line53-55)
Comment: 3. Line 64-66 needs a reference.
Response: Thank you for your valuable suggestions on this article. We have added references on lines 67.
1.Smolander, M.; Alakomi, H.L.; Ritvanen, T.; Vainionpaa, J.; Ahvenainen, R. Monitoring of the quality of modified atmosphere packaged broiler chicken cuts stored in different temperature conditions. A. Time-temperature indicators as quality-indicating tools. Food Control 2004, 15, 217-229, doi:10.1016/s0956-7135(03)00061-6.
- Zhang, Q.Q.; Han, Y.Q.; Cao, J.X.; Xu, X.L.; Zhou, G.H.; Zhang, W.Y. The spoilage of air-packaged broiler meat during storage at normal and fluctuating storage temperatures. Poultry Science 2012, 91, 208-214, doi:10.3382/ps.2011-01519.
Comment: 4. In the last paragraph of the Introduction the Ratkowsky model deserves to be mentioned as well.
Response: Thank you for your valuable suggestions. which have been supplemented in the section "1. introduction" in this article. (Line 86-88)
Comment: 5. Line 134. Replace ‘MES’ by ‘MSE’ in the equation.
Response: Sorry for the error. It has been corrected. (Line 142)
Comment: 6. Line 158: lg10 is not the ‘natural logarithm’.
Response: The error has been corrected. (Line 166)
Comment: 7. Overall comment ‘Results and Discussion’ section. When reading this section, it gets the impression that the ‘Results section’ and the ‘Discussion section’ can best be separated. In the current state the flow of the results in the different parts is disturbed by a discussion until the next results part starts. Each result section is a standalone and when combined will be more easily readable. It also prevents any repeats in the various discussion parts.
Secondly, this whole section can be written much more concisely. There are too many repeats, and many details are mentioned in the text that are already given in the figures.
Therefore, I advise the authors to improve this part of their manuscript.
Response: Thank you for your valuable opinions on this project. Appropriate pruning has been made to the Results section to make the content more concise.
Comment: 8. Figures 1-4. Make sure that all y-axes are the same in each figure. For instance, in Figure 1 the second top panel (Ps. ludensis at 4°C) the Y-axis should also run to 10, like in the other panels.
Response: Thanks for your valuable comments. We have made modifications to the y-axis of Figures 1-4.
Comment: 9. Line 257: Write MDA full out as it is used here for the first time.
Explain here as well whether lipid oxidation is a microbial process or can it also be chemical oxidation in the presence of oxygen?
Response: Thanks for your valuable comments, the full MDA name has been added. Additionally, lipid oxidation can be chemical oxidation in the presence of oxygen and lipoxygenase. We have added this part to Section 3.2.2 (line260-262,264).
Comment: 10. Line 356 Replace ‘weakened’ by ‘was less effective’.
Response: Thanks for your valuable suggestion. Modifications have been made as suggested (line371).
Comment: 11. Line 395 Typo, should be ‘provided’.
Response: Sorry for the mistake, the spelling on line 395 has been corrected(line410).
Reviewer 3 Report
The present manuscript deals with combination of modified atmosphere packaging and cold storage strategy for preservation of chicken. Research methodology has been designed appropriately and results have been present well with proper justification. However, introduction needs to be improved. I recommend this article for publication with minor revision.
1. Some of the relevant studies by other researchers on use of MAP and low temperature need to be cited in the introduction.
2. Line 95: “sterilized by irradiation treatment”, which irradiation method? UV or gamma or any other? Kindly specify.
3. Line 96: “3–4 lg CFU/g”, lg is the correct abbreviation of log? Log is already an abbreviation of logarithm. Kindly check and correct throughout.
4. Which packaging material was used for both normal packaging and modified atmosphere packaging?
5. On the basis of growth of microorganisms, shelf life of chicken packed in both packages and stored at different temperature shall be determined. Authors should include one section about the shelf life under Results and discussion.
6. Grammatical errors need to be corrected throughout the manuscript.
Author Response
Response to the reviewer #3’s comments:
Comment: 1 Some of the relevant studies by other researchers on use of MAP and low temperature need to be cited in the introduction.
Response: Thank you for your valuable suggestions. which have been supplemented in the section "1. introduction" in this article. (Line57-61)
1.Guerra Monteiro, M.L.; Marsico, E.T.; Teixeira, C.E.; Mano, S.B.; Conte Junior, C.A.; Vital, H.d.C. Shelf life of refrigerated tilapia fillets (Oreochromis niloticus) packed in modified atmosphere and irradiated. Ciencia Rural 2012, 42, 737-743, doi:10.1590/s0103-84782012000400027.
2.Jimenez, S.M.; Salsi, M.S.; Tiburzi, M.C.; Rafaghelli, R.C.; Tessi, M.A.; Coutaz, V.R. Spoilage microflora in fresh chicken breast stored at 4 degrees C: influence of packaging methods. Journal of applied microbiology 1997, 83, 613-618, doi:10.1046/j.1365-2672.1997.00276.x.
Comment: 2、Line 95: “sterilized by irradiation treatment”, which irradiation method? UV or gamma or any other? Kindly specify.
Response: Thank you for your valuable suggestions on this article. We have been added to the section "2.2 Sample preparation and inoculation" in this article. (Line101-102)
Comment: 3、 Line 96: “3–4 lg CFU/g”, lg is the correct abbreviation of log? Log is already an abbreviation of logarithm. Kindly check and correct throughout.
Response: First of all, thank you for your question. log is a logarithmic function, and lg is a base-10 logarithmic function.
Comment: 4、Which packaging material was used for both normal packaging and modified atmosphere packaging?
Response: Thanks for your valuable comments, polypropylene film was used and we have supplemented.
Comment: 5、 On the basis of growth of microorganisms, shelf life of chicken packed in both packages and stored at different temperature shall be determined. Authors should include one section about the shelf life under Results and discussion.
Response: Thanks for your valuable advice. The shelf life of chicken packed in both packages has been supplemented. (Lines 313-321)
Comment: 6、Grammatical errors need to be corrected throughout the manuscript.
Response: Sorry for the mistake, the grammar of the full text has been corrected.